# Impact of body composition on patient prognosis after SARS-Cov-2 infection

**Takayuki Yamamoto[1], Kazushi Sugimoto[2], Syuhei Ichikawa[1], Kei Suzuki[3], Hideki Wakabayashi[1], Kaoru Dohi [4], Norihiko Yamamoto [1]***

**1** Department of General Medicine, Mie University Hospital, Tsu, Japan, **2** Department of Clinical Laboratory, Mie University Hospital, Tsu, Japan, **3** Department of Infectious Disease, Mie University Hospital, Tsu, Japan, **4** Department of Cardiology and Nephrology, Mie University Hospital, Tsu, Japan

\* kotetsu@med.mie-u.ac.jp

**Data Availability Statement:** All relevant data are within the paper and its Supporting Information files.

## Abstract

### Background

Since the first outbreak of coronavirus disease 2019 (COVID-19), it has been reported that several factors, including hypertension, type 2 diabetes mellitus, and obesity, have close relationships with a severe clinical course. However, the relationship between body composition and the prognosis of COVID-19 has not yet been fully studied.

### Methods

The present study enrolled 76 consecutive COVID-19 patients with computed tomography (CT) scans from the chest to the pelvis at admission. The patients who needed intubation and mechanical ventilation were defined as severe cases. Patients were categorized into four groups according to their body mass index (BMI). The degree of hepatic steatosis was estimated by the liver/spleen (L/S) ratio of the CT values. Visceral fat area (VFA), psoas muscle area (PMA), psoas muscle mass index (PMI), and intra-muscular adipose tissue content (IMAC) were measured by CT scan tracing. These parameters were compared between non-severe and severe cases.

### Results

Severe patients had significantly higher body weight, higher BMI, and greater VFA than non-severe patients. However, these parameters did not have an effect on disease mortality. Furthermore, severe cases had higher IMAC than non-severe cases in the non-obese group.

### Conclusions

Our data suggest high IMAC can be a useful predictor for severe disease courses of COVID-19 in non-obese Japanese patients, however, it does not predict either disease severity in obese patients or mortality in any obesity grade.

1 / 13

**Funding:** The author(s) received no specific funding for this work.

**Competing interests:** The authors have declared that no competing interests exist.

**Abbreviations:** COVID-19, coronavirus disease 2019; BMI, body mass index; VFA, visceral fat area; PMA, psoas muscle area; PMI, psoas muscle mass index; IMAC, intra-muscular adipose tissue content; SARS-Cov-2, Respiratory Syndrome-Coronavirus-2.

## Introduction

Since the first outbreak in Wuhan, China in 2019, coronavirus disease 2019 (COVID-19), which is caused by Severe Acute Respiratory Syndrome-Coronavirus-2 (SARS-Cov-2), has spread worldwide. While most cases of other coronavirus infections were acute and cleared quickly, COVID-19 has sometimes been fatal, with a mortality rate of around 2% initially [1], and the most common cause of death was acute respiratory distress syndrome (ARDS).

Factors that can predict disease severity and worse outcomes in COVID-19 patients have been investigated in many studies. In these studies, it was suggested that several risk factors were associated with worse disease outcomes, including high age [2], type 2 diabetes mellitus (T2DM) [3], and cardiovascular disease (CVD) [4]. In addition, many studies, including a meta-analysis, showed the COVID-19 patients with obesity had a worse disease course [5–7]. For example, Zhang *et al.* analyzed 22 cohort studies from seven countries performed in the first six months of the pandemic and showed that obese patients had more severe disease than non-obese patients and were more likely to develop ARDS or need invasive mechanical ventilation, but obesity *per se*. was not associated with the mortality of COVID-19 [6]. Another study by Chu *et al.* also found that, though obesity was a risk factor for severe COVID-19, especially in younger patients, it did not increase the risk of hospital mortality [5].

In most studies that linked obesity and COVID-19 outcome, obesity was assessed by body mass index (BMI). However, using the BMI to determine obesity has several problems. First, the definition of obesity is different across countries. For instance, the diagnostic criterion for obesity is BMI $\geq$ 30 kg/m$^2$ in the United States and European countries, but patients with BMI $\geq$ 25 kg/m$^2$ are diagnosed as obese in Japan. In fact, two previous meta-analyses by Zhang and Wang included studies that had different BMI cut-offs to define obesity. Second, because BMI is calculated only by body weight and height, it sometimes does not reflect body composition precisely. A certain number of individuals have excess body fat despite a normal BMI, and these individuals usually have excess visceral fat, which leads to insulin resistance, T2DM, hypertension, and metabolic syndrome, as well as obese individuals. Therefore, we may miss the patients who potentially have the same risks as obese patients by assessing BMI alone. For this reason, Stevanovic *et al.* examined the effects of body and visceral fat mass on the course and outcome of COVID-19 in Serbian patients, and they found that these parameters were stronger predictors of outcome than BMI alone [8]. Furthermore, it has recently been reported that body composition indices, such as muscle mass or degree of muscular fatty degeneration, are also related to worse prognoses of several diseases, including malignant disease and acute illness [9–12]. However, the relationship with the outcome of COVID-19 has not yet been fully clarified.

Taking these findings from previous studies into account, in the current study, the impacts of body composition indices, including the psoas muscle mass index (PMI) and intra-muscular adipose tissue content (IMAC), which reflect muscle quality and are highly correlated with sarcopenia, on the clinical course of COVID-19 were investigated in Japanese patients.

## Methods

### Study population

Patients consecutively hospitalized for COVID-19 at Mie University Hospital from August 2020 to September 2021 were enrolled. The patients were followed, and clinical parameters were obtained during hospitalization. Those who needed intubation and mechanical ventilation were defined as severe cases. The indication for intubation and mechanical ventilation followed the "Clinical Management of Patients with COVID-19: A guide for front-line healthcare

workers" by the Ministry of Health, Labour and Welfare, Japan. Briefly, the patients were intubated and on mechanical ventilation when their blood oxygen saturation became $\leq 93\%$ even if administered high-flow nasal oxygen. The patients were excluded from further analysis if clear computed tomography (CT) images at admission to perform anthropometric measurements were not obtained. Informed consent was obtained from each patient enrolled in the study in writing or orally. This study was approved by the Ethics Committee of Mie University Hospital (reference number H2022-206), ensuring that it conformed to the ethical guidelines of the 2013 Declaration of Helsinki.

## Data collection

Physical and routine laboratory data (complete blood count, biochemistry, coagulation parameters) at admission were obtained. In addition, CT from the chest to the pelvis was also taken for each patient at admission. All the blood tests were performed at the clinical laboratory of Mie University Hospital using LABOSPECT 008 (Hitachi High-Tech Corporation, Tokyo, Japan) for blood chemistry, XN-3000 (Sysmex, Kobe, Japan) for complete blood count, and CS-5100 (Sysmex) for blood coagulation tests. CT scans were performed using Discovery CT750 HD (GE Health Care, Tokyo, Japan).

BMI was calculated using the formula: BMI $(kg/m^2)$ = body weight (kg)/(body height $(m))^2$. Obese patients were categorized into four groups by BMI according to the Guidelines for the Management of Obesity Disease 2022 by Japanese Society for the Study of Obesity as follows: $25 \leq BMI < 30$ as obesity grade **1**; $30 \leq BMI < 35$ as obesity grade **2**; $35 \leq BMI < 40$ as obesity grade **3**; $BMI \geq 40 \ kg/m^2$ as obesity grade **4**. Patients with $BMI < 25 \ kg/m^2$ were also categorized as obesity grade **0**.

The degree of hepatic steatosis was estimated by the liver/spleen (L/S) ratio of the CT values. Both hepatic and splenic CT values were measured using three circular regions of interest with 2-cm diameters. An L/S ratio < 1.1 was considered to indicate hepatic steatosis. Visceral fat area (VFA) $(cm^2)$ and psoas muscle area (PMA) $(cm^2)$ were measured by tracing the CT image at the umbilical level. Skeletal muscle mass was assessed by PMI $(cm^2/m^2)$ calculated by the formula: PMA $(cm^2)/(height \ (m))^2$. Skeletal muscle quality was evaluated by IMAC, which was calculated by the CT value ratio of the multifidus muscle/back subcutaneous adipose tissue at the umbilical level. All anthropometric measurements on CT were performed using EV Insite software (PSP Corporation, Tokyo, Japan).

## Statistical analysis

All statistical analyses were conducted using R 4.1.2 on RStudio build 554, with the following packages: tidyverse, readxl, ggplot2, patchwork, and tableone. The Chi-square test was used to test categorical data, and the Mann-Whitney U test was used to test continuous data. P values less than 0.05 were considered significant. All analyses were performed by a professional biostatistician.

## Results

### Patients' characteristics

Overall, 93 COVID-19 patients were admitted to Mie University Hospital during the study period. Of them, 17 were excluded because they lacked CT images at admission (eight patients were pregnant, required area were not captured in eight patients, clear image was not obtained in one patient), and 76 were further analyzed for the study. The flowchart of the patient enrollment is shown in Fig 1 and the characteristics of the 76 patients at admission are shown in

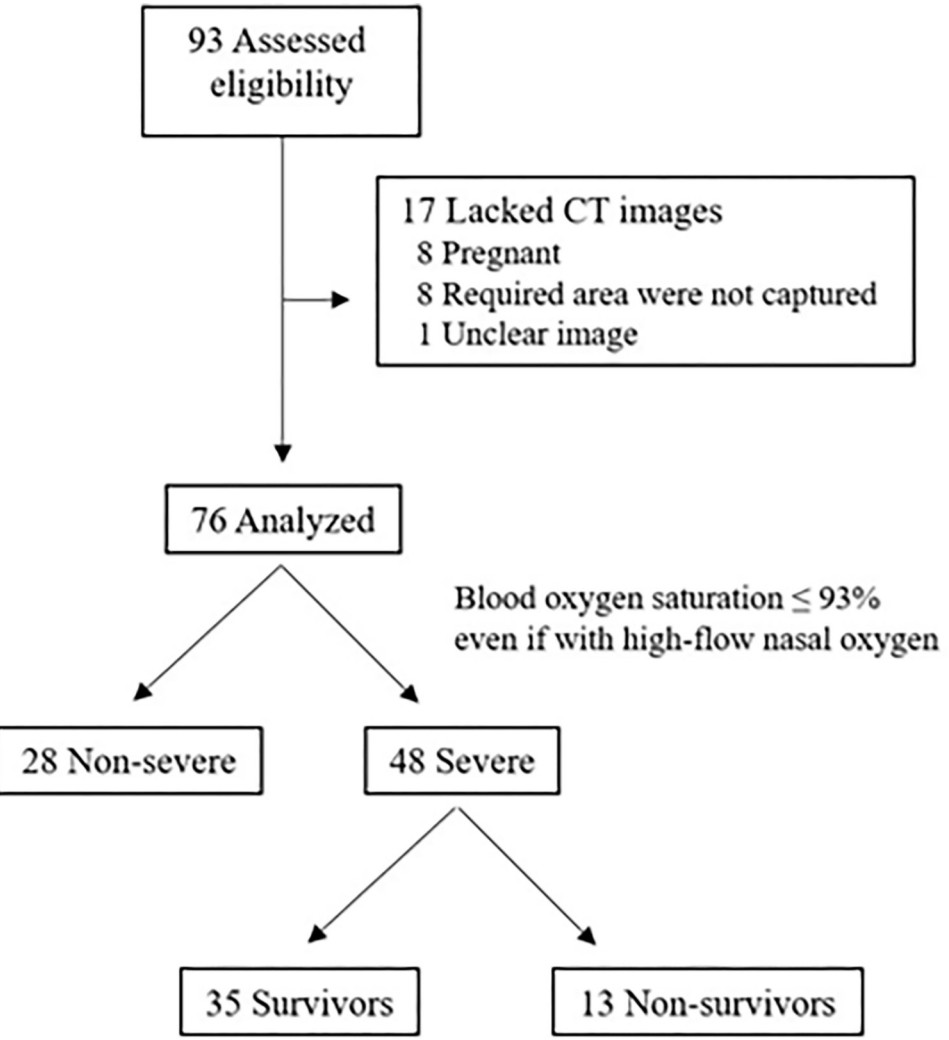

**Fig 1. Flowchart of the patient enrollment.** Overall, 93 COVID-19 patients were admitted to Mie University Hospital during the study period. Of them, 17 were excluded because they lacked CT images at admission and 76 were further analyzed for the study. Among them, 48 patients became severe and 13 patients died.

Table 1. Fifty-four (70%) were male, their median age was 59 years, and 28 patients (37%) had T2DM. The median weight was 71.5 kg, and median BMI was 26.8 kg/m². Twenty-four patients (31.6%) were categorized as obesity grade 0, 28 (36.8%) were categorized as obesity grade 1, and 24 patients were categorized as obesity grade $\geq$ 2. In addition, 45 patients (59.2%) had L/S ratios less than 1.1, indicating that these patients had liver steatosis.

## Comparison of clinical parameters between non-severe and severe cases

Of the 76 enrolled patients, 28 did not need intubation and were categorized as the non-severe cases, and 48 patients needed intubation and mechanical ventilation and were categorized as the severe cases. As the next step, non-severe and severe cases were compared. As shown in Table 2, severe cases had significantly higher serum AST, ALT, and LDH values, reflecting more severe organ damage. Moreover, severe cases had significantly higher CRP and WBC levels, reflecting more active inflammation and a thrombotic tendency. Contrary to a previous report [3], T2DM was not related to a worse disease course in the present cohort.

**Table 1. Characteristics of the 76 enrolled patients at admission.**

|  |  | n = 76 |
|---|---|---|
| **Age (y)** |  | 59 (22–85) |
| **Sex (M/F)** |  | 54/22 |
| **T2DM** |  | 28 (36.8%) |
| **AST (IU/L)** |  | 57 (13–477) |
| **ALT (IU/L)** |  | 42 (10–205) |
| **LDH (IU/L)** |  | 461 (183–1729) |
| **Creatinine (mg/dL)** |  | 0.82 (0.15–16) |
| **CRP (mg/dL)** |  | 9.95 (0.2–29.38) |
| **D-dimer (ng/mL)** |  | 1.20 (0.1–84.1) |
| **WBCs (/μL)** |  | 7170 (1400–24660) |
| **Lymphocytes (/μL)** |  | 690 (216–3060) |
| **Platelets (x10³/μL)** |  | 183 (32–484) |
| **L/S ratio** |  | 0.96 (0.1–1.8) |
| **L/S ratio < 1.1** |  | 45 (59.2%) |
| **Weight (kg)** |  | 71.5 (41–160) |
| **BMI (kg/m²)** |  | 26.8 (17–58.6) |
| **Obesity grade** |  |  |
|  | **0** | 24 |
|  | **1** | 28 |
|  | **2** | 13 |
|  | **3** | 6 |
|  | **4** | 5 |
| **VFA (cm²)** |  | 128.7 (13.6–419.5) |
| **PMA (cm²)** |  | 17.9 (3.3–37.1) |
| **PMI** |  | 6.72 (1.12–15.21) |
| **IMAC** |  | -0.33 (-0.73–0.02) |

T2DM, type 2 diabetes mellitus; AST, aspartate aminotransferase; ALT, alanine aminotransferase; LDH, lactate dehydrogenase; CRP, C-reactive protein; WBCs, white blood cells; L/S ratio, liver/spleen ratio; BMI, body mass index; VFA, visceral fat area; PMA, psoas muscle area; PMI, psoas muscle mass index; IMAC, intra-muscular adipose tissue content.

Data are expressed as medians and ranges for continuous variables, and numbers and percentages for categorical variables.

Differences in the parameters for body composition were compared between the two groups, and body weight, BMI, and degree of obesity were significantly higher in severe cases. In addition, severe cases also had greater VFA, but there were no significant differences in the L/S ratio, PMI, and IMAC between these two groups.

## Comparison of clinical parameters between survivors and non-survivors

In total, 13 of 28 severe cases died. The differences between surviving and non-surviving severe cases were examined. As shown in Table 3, the serum creatinine level was higher and the platelet count was lower in non-survivors. On the other hand, there were no differences in other blood tests and body composition parameters including BMI, obesity degree, and VFA, although these parameters differed between non-severe and severe cases (**Table 2**).

**Table 2. Comparison between non-severe and severe cases.**

| | Non-severe (n = 28) | Severe (n = 48) | p-values |
|---|---|---|---|
| Age (y) | 59 (22–85) | 58 (40–79) | 0.428 |
| Sex (M/F) | 19/9 | 35/13 | 0.836 |
| T2DM | 10 (35.7%) | 18 (37.5%) | 1.000 |
| AST (IU/L) | 43 (13–132) | 64 (21–477) | 0.022 |
| ALT (IU/L) | 28 (10–96) | 48 (15–205) | 0.010 |
| LDH (IU/L) | 388 (183–1155) | 524 (234–1729) | 0.003 |
| Creatinine (mg/dL) | 0.89 (0.15–16) | 0.78 (0.35–13.6) | 0.386 |
| CRP (mg/dL) | 6.70 (0.02–20) | 12.02 (0.2–29.38) | 0.022 |
| D-dimer (ng/mL) | 0.92 (0.1–3.89) | 1.22 (0.1–84.21) | 0.088 |
| WBCs (/μL) | 5310 (1400–20960) | 7730 (2930–24660) | 0.012 |
| Lymphocytes (/μL) | 745 (280–3060) | 640 (216–2660) | 0.613 |
| Platelets (x$10^3$/μL) | 167 (32–402) | 200 (42–484) | 0.175 |
| L/S ratio | 0.97 (-0.1–1.2) | 0.93 (0.4–1.8) | 0.349 |
| L/S ratio < 1.1 | 16 (57.1%) | 29 (60.4%) | 0.970 |
| Weight (kg) | 67.0 (45–111) | 76.0 (41–160) | 0.002 |
| BMI (kg/m$^2$) | 24.0 (18.9–43) | 27.7 (17–58.6) | 0.001 |
| Obesity grade | | | 0.009 |
| 0 | 16 | 8 | |
| 1 | 7 | 21 | |
| 2 | 3 | 10 | |
| 3 | 1 | 5 | |
| 4 | 1 | 4 | |
| VFA (cm$^2$) | 111.7 (13.6–236.9) | 159.0 (19.9–419.5) | 0.003 |
| PMA (cm$^2$) | 18.3 (5.95–29.8) | 17.7 (3.3–38.9) | 0.357 |
| PMI | 6.61 (2.59–10.3) | 6.73 (1.120–15.22) | 0.579 |
| IMAC | -0.33 (-0.73–0.03) | -0.33 (-0.5–0.02) | 0.343 |

T2DM, type 2 diabetes mellitus; AST, aspartate aminotransferase; ALT, alanine aminotransferase; LDH, lactate dehydrogenase; CRP, C-reactive protein; WBCs, white blood cells; L/S ratio L/S ratio, liver/spleen ratio; BMI, body mass index; VFA, visceral fat area; PMA, psoas muscle area; PMI, psoas muscle mass index; IMAC, intra-muscular adipose tissue content.

Data are expressed as medians and ranges for continuous variables, and numbers and percentages for categorical variables. The Chi-squared test was used to test categorical data, and the Mann-Whitney U test was used to test continuous data.

## Effect of body composition on disease severity according to degree of obesity

Finally, a subanalysis was performed to determine how body composition affects disease severity in non-obese (obesity grade **0**), mildly obese (obesity grade **1**), and moderately to severely obese (obesity grades **2–4**) patients. During hospitalization, eight of 24 non-obese cases, 21 of 28 mildly obese cases, and 19 of 24 moderately to severely obese cases developed severe disease. Table 4 shows the comparison of the age between non-severe and severe cases in each obesity group. Non-severe patients were younger than severe cases in obesity grade 0 group, while there was no difference of age between non-severe and severe cases in obesity grade 1 and 2–4 grade groups. Fig 2 shows the comparisons of body composition parameters between non-severe and severe cases in each obesity group. In the obesity grade 0 group, severe cases had

**Table 3. Comparison between survivors and non-survivors in severe patients.**

|  | | Survivors (n = 35) | Non-survivors (n = 13) | p-values |
|---|---|---|---|---|
| Age (y) | | 57 (32–83) | 63 (40–77) | 0.097 |
| Sex (M/F) | | 25/10 | 10/3 | 0.988 |
| T2DM | | 13 (37.1%) | 5 (38.5%) | 1.000 |
| AST (IU/L) | | 64 (20–151) | 54 (19–108) | 0.219 |
| ALT (IU/L) | | 53 (15–110) | 32 (17–205) | 0.054 |
| LDH (IU/L) | | 537 (256–1042) | 409 (234–1729) | 0.150 |
| Creatinine (mg/dL) | | 0.73 (0.35–7.81) | 1.41 (0.44–13.6) | 0.016 |
| CRP (mg/dL) | | 12.24 (0.2–23.87) | 10.26 (0.76–29.38) | 0.668 |
| D-dimer (ng/mL) | | 1.18 (0.1–22.53) | 1.78 (0.3–84.21) | 0.219 |
| WBCs (/μL) | | 7890 (3420–22340) | 6780 (2930–24660) | 0.410 |
| Lymphocytes (/μL) | | 630 (100–2660) | 750 (310–2200) | 0.531 |
| Platelets (x10$^3$/μL) | | 214 (42–272) | 139 (50–484) | 0.007 |
| L/S ratio | | 0.93 (-0.1–1.4) | 0.97 (0.2–1.3) | 0.862 |
| L/S ratio < 1.1 | | 21 (60.0%) | 8 (61.5%) | 1.000 |
| Body Weight (kg) | | 75.0 (41–160) | 77.0 (47–150) | 0.935 |
| BMI (kg/m$^2$) | | 28.5 (17–49) | 26.9 (17–58.6) | 0.531 |
| Obesity grade | | | | 0.935 |
| | 0 | 6 | 2 | |
| | 1 | 14 | 7 | |
| | 2 | 8 | 2 | |
| | 3 | 4 | 1 | |
| | 4 | 3 | 1 | |
| VFA (cm$^2$) | | 155.2 (19.9–419.5) | 167.8 (88–270) | 0.991 |
| PMA (cm$^2$) | | 17.9 (5.9–38.9) | 16.8 (3.3–3.71) | 0.862 |
| PMI | | 6.77 (3.12–15.22) | 5.95 (1.12–14.30) | 0.898 |
| IMAC | | -0.33 (-0.52–0.03) | -0.35 (-0.49–0.02) | 0.736 |

T2DM, type 2 diabetes mellitus; AST, aspartate aminotransferase; ALT, alanine aminotransferase; LDH, lactate dehydrogenase; CRP, C-reactive protein; WBC, white blood cells; L/S ratio L/S ratio, liver/spleen ratio; BMI, body mass index; VFA, visceral fat area; PMA, psoas muscle area; PMI, psoas muscle mass index; IMAC, intra-muscular adipose tissue content.

Data are expressed as medians and ranges for continuous variables, and numbers and percentages for categorical variables. The Chi-squared test was used to test categorical data, and the Mann-Whitney U test was used to test continuous data.

significantly higher IMAC than non-severe cases, which means that these patients had more fat degeneration in their skeletal muscles, whereas no differences were found in PMI, L/S ratio, and VFA between non-severe and severe cases. Moreover, no body composition parameter was different between non-severe and severe cases in any other obesity grade. In addition,

**Table 4. Comparison of age between non-severe and severe cases.**

|  | Non-severe (n = 28) | Severe (n = 48) | p-values |
|---|---|---|---|
| Obesity grade 0 | 59.5 (38–81) | 63 (40–77) | 0.015 |
| Obesity grade 1 | 60 (50–83) | 58 (29–85) | 0.633 |
| Obesity grade 2–4 | 50 (38–64) | 49 (22–71) | 0.594 |

Data are expressed as medians and ranges. The Mann-Whitney U test was used for comparison.

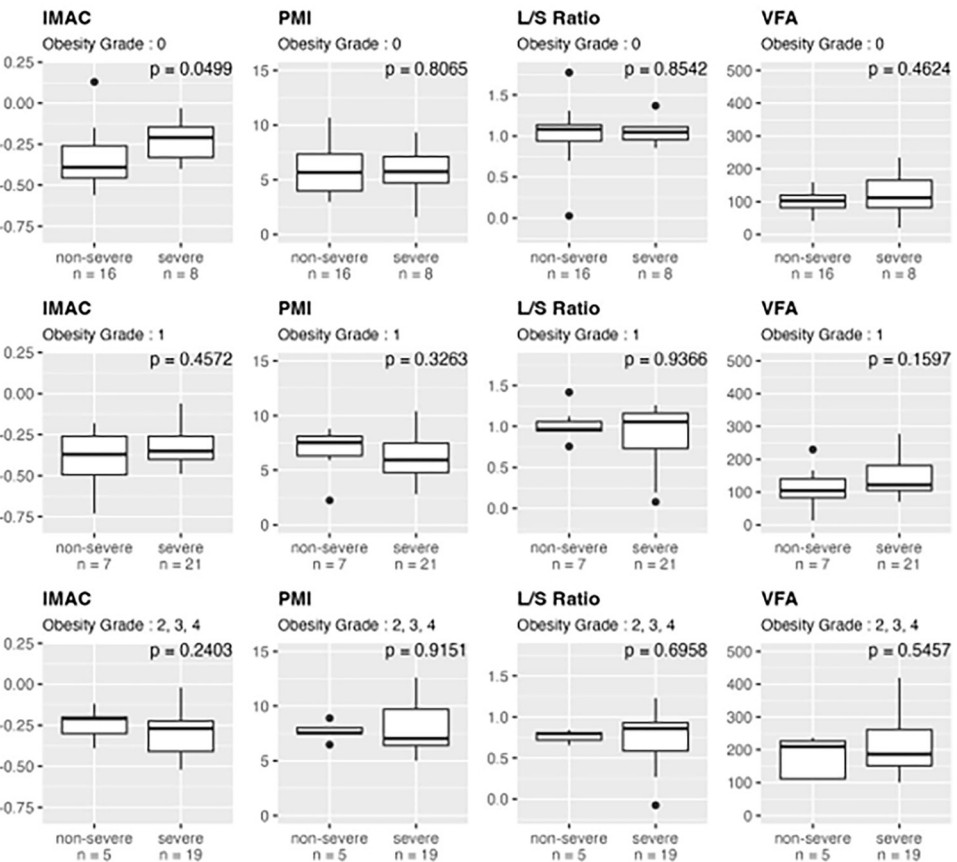

**Fig 2. Comparison of the body composition parameters between non-severe and severe cases in each obesity grade.** Severe cases have significantly higher IMAC compared to non-severe cases in the obesity grade 0 group, whereas there are no differences in PMI, L/S ratio, and VFA. Moreover, no body composition parameter is different between non-severe and severe cases in any other obesity grade.

there was no correlation between IMAC and disease mortality in any obesity group and in the overall analysis (Tables 3 and S1).

## Discussion

In this study, higher body weight, higher BMI and obesity grade, and greater VFA were related to more severe COVID-19. However, these parameters did not affect disease mortality. In addition, severe cases had higher IMAC than non-severe cases in the non-obese group.

It has been reported from many countries other than Japan that obesity and increased BMI are major risk factors for more severe disease after SARS-Cov-2 infection [5–7,13]. These studies' results were obtained mainly in Western countries or China. In addition, early studies in Japan also showed that obesity was a risk factor for severity on admission for COVID-19, as well as older age, CVD, or T2DM [14]. The present results were compatible with these previous reports.

The current study also showed that severe patients had higher VFA. There are several reasons why obesity, especially visceral obesity, can be linked to many kinds of disease. One explanation is that obesity is often associated with T2DM, and their association causes a dysfunctional immune response, reduced pulmonary function, thrombogenic potential, increased cardio-renal stress, etc., which lead to increased severity of many diseases including

COVID-19 [7]. In addition, visceral adipose tissue is now thought to be the largest endocrine organ in the human body. It has been shown that visceral adipocytes secrete several cytokines, named adipokines, including leptin and adiponectin, which regulate appetite and body weight. In addition, they can also secrete inflammatory cytokines such as TNF-$\alpha$ and IL-6, and as visceral adipocytes grow bloated, they come to produce greater amounts of inflammatory cytokines and less adiponectin [15]. In fact, Werida et al. reported that circulating IL-6 levels were significantly elevated in obese subjects of both sexes [16]. Subsequently, these inflammatory cytokines exacerbate systemic inflammation and lead to the development and progression of many diseases.

Although a growing body of evidence suggests that obesity increases the risk for a worse disease course, the relationship between obesity and mortality of COVID-19 is still controversial. For example, Popkin et al. reported that individuals with obesity had a 48% increase in death [17]. Furthermore, Albarran-Sanchez showed that the mortality rates of hospitalized patients whose BMI was 35–39.9 kg/m$^2$ and > 40 kg/m$^2$ increased to 50% and 58.8%, respectively, in contrast to patients with normal weight who had a mortality rate of 43.3% [18]. However, other studies did not find an association between obesity and increased risk of death from COVID-19 [5,6,19]. In the current study, the data also did not show a relationship between obesity and disease mortality. It still remains unclear why obesity was not associated with COVID-19 mortality in many previous studies and the present study, though obesity has many negative effects on the disease course. Therefore, more study is needed.

According to the National Health and Nutrition Survey in 2019 by the Ministry of Health, Labour and Welfare, Japan, the prevalence of obesity (BMI $\geq$ 25 kg/m$^2$) in Japanese males and females was 33% and 22%, respectively (https://www.mhlw.go.jp/content/000710991.pdf), and the majority of the Japanese population have normal weight. Moreover, it is difficult to simply compare study data in Japan to those from other countries or combine them, because Japan has a different definition for obesity from Western countries. Therefore, another predictor for COVID-19 severity was needed, especially in non-obese subjects. In fact, eight of 24 patients in the non-obese group had severe disease courses in the present study.

Recently, it has been shown that sarcopenia is related to the development or worse prognosis of many diseases, including the severity of COVID-19 [20–22]. Sarcopenia develops when the amount of muscle breakdown exceeds the amount of muscle production due to changes in hormones and inflammatory cytokines involved in muscle metabolism. Sarcopenia can also be associated with obesity when low muscle mass and increased adiposity are present in obese individuals, and it increases mortality [23,24]. In the pathogenesis of sarcopenic obesity, inflammation of adipose tissues and muscles caused by adipokines and myokines plays an important role [25], and sarcopenic obesity can have more harmful effects on COVID-19 outcomes than obesity alone. Nevertheless, the present study results showed that neither PMA nor PMI, which indicate the degree of sarcopenia, were associated with disease severity or mortality in any obesity grade.

On the other hand, IMAC directly reflects adipose degeneration and the quality of muscle and can thus be a more precise predictor of sarcopenia than measuring muscle mass alone. IMAC has also been reported to be related to poor prognosis of several diseases, such as metabolic diseases and cancers [9,26–29]. However, how IMAC affects the prognosis after SARS-Cov-2 infection has not yet been clarified. Therefore, one of the most important findings of the current study is that higher IMAC was related to a worse prognosis of COVID-19 even in non-obese subjects, whereas other markers for sarcopenia (PMA, PMI) had no relationship with disease outcomes in this patient group.

Though the precise mechanism regarding the association between IMAC and disease severity of COVID-19 is still unclear, there are some speculations. Recent study by Yi et al. showed

thoracic myosteatosis was related to progression of COVID-19 [30], which is comparable with our results. They speculate that muscle adiposity may cause systemic muscle weakness and diminished effort for coughing which is crucial for protection against respiratory infection, and can leads to disease progression. The other possible mechanism is the changes in myokines. For example, Addison et al. showed frail patients had higher muscle adiposity and these muscle tissues with fatty degeneration produced increased level of IL-6 [31]. IL-6 induces anti-inflammatory and regenerative responses via classic signaling to restricted cells such as hepatocytes or leucocytes, while its trans-signaling induces pro-inflammatory responses to all cells in the body [32]. In muscle cells, IL-6 signaling is associated with myogenesis and muscle growth, however, it also causes muscle atrophy and wasting [33]. Furthermore, Kang et al. reported IL-6 trans-signaling plays a crucial role for the pathogenesis of cytokine release syndrome in COVI-19 [34]. Therefore, it is possible that IL-6 produced by muscle tissue with fatty degeneration affect disease course of COVID-19 to some extent. However, these effects by myokines may become less remarkable as the patient gets obese, because the effects of the larger amount of inflammatory adipokines secreted by the visceral adipose tissue will be more prominent. Our data also showed non-severe patients were younger than severe patients only in obesity grade 0 group. As non-obese individuals are less likely to have other metabolic-associated risk factors, it is possible that high IMAC may be more crucial for COVID-19 severity in this group and non-obese younger patients had more favorable disease courses because they had less muscle adiposity compared to non-obese older patients.

Among the limitations of the current study was the relatively small number of subjects because it was conducted at a single facility; nevertheless, it made the study results more reliable, since the definition of disease severity, therapeutic methods, and protocol of the data collection were standardized. In addition, whether IMAC was associated with disease mortality could not be determined in this study due to the low number of fatal outcomes in each obesity group; therefore, further evaluation with a larger number of patients is needed to confirm the present findings. Furthermore, it is also important to confirm whether IMAC can be a predictor for a worse disease course in non-obese individuals in other countries whose definition of obesity (BMI $\geq$ 30 kg/m$^2$) differs from that of Japan. Therefore, further studies are necessary.

In conclusion, higher BMI and visceral obesity are associated with a worse prognosis in COVID-19 patients. In addition, higher IMAC is also related to poor disease outcomes in non-obese patients. In this study period, the majority of COVID-19 infections were caused by the delta strain, which had much higher disease severity and mortality than the later omicron strain. However, it remains possible that the coronavirus may again mutate to a virulent variant after the current pandemic comes to an end, or an outbreak of other lethal viruses may occur in the near future. In our view, IMAC will potentially be a useful predictor of disease courses in such situations.

## Supporting information

**S1 Table. Comparison of IMAC between survivors and non-survivors in each obesity grade in severe patients.**
(DOCX)

## Author Contributions

**Conceptualization:** Kazushi Sugimoto, Norihiko Yamamoto.

**Data curation:** Takayuki Yamamoto, Kazushi Sugimoto, Syuhei Ichikawa, Kei Suzuki.

**Formal analysis:** Takayuki Yamamoto, Syuhei Ichikawa.

**Investigation:** Kazushi Sugimoto, Syuhei Ichikawa, Norihiko Yamamoto.

**Methodology:** Kazushi Sugimoto, Syuhei Ichikawa, Norihiko Yamamoto.

**Project administration:** Takayuki Yamamoto, Kazushi Sugimoto, Norihiko Yamamoto.

**Resources:** Takayuki Yamamoto, Kei Suzuki.

**Software:** Syuhei Ichikawa.

**Supervision:** Kazushi Sugimoto, Hideki Wakabayashi, Kaoru Dohi, Norihiko Yamamoto.

**Validation:** Kazushi Sugimoto, Syuhei Ichikawa.

**Visualization:** Takayuki Yamamoto, Kazushi Sugimoto, Norihiko Yamamoto.

**Writing – original draft:** Takayuki Yamamoto.

**Writing – review & editing:** Kazushi Sugimoto, Norihiko Yamamoto.

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
