## [Decision Letter · Decision Letter 0]

23 Feb 2023

PONE-D-23-01341Impact of body composition on patient prognosis after SARS-Cov-2 infectionPLOS ONE

Dear Dr. Yamamoto,

Thank you for submitting your manuscript to PLOS ONE. After careful consideration, we feel that it has merit but does not fully meet PLOS ONE’s publication criteria as it currently stands. Therefore, we invite you to submit a revised version of the manuscript that addresses the points raised during the review process.

We look forward to receiving your revised manuscript.

Kind regards,

Jun Mori

Academic Editor

PLOS ONE

Journal Requirements:

**Additional Editor Comments:**

The authors assess the body composition and the degree of sarcopenia between severe and non-severe patients with COVID-19. They concluded that Intra-muscular adipose tissue content (IMAC) could be a useful predictor of disease severity for patients with COVID-19. The data is somewhat interesting, but I have the following concerns.

1. Non-severe patients group includes young patient such as 20 years age, even though mean age was not different between severe and non-severe groups. In general, younger people tend to be less severe with COVID-19 infection compared to older people. I'm wondering why this young patient was admitted and CT scan was performed. You should show the patient's age for both groups in detail.

2. There are many risk factors of severity of COVID-19 other than obesity and T2DM, e.g. hypertension, hyperlipidemia, chronic lung disease, chronic kidney disease, tobacco habits. What are your thoughts on the involvement of these other risk factors?

3. Do you have the data of waist circumference, which is also an index of visceral fat mass? Measuring waist circumference is much easier to evaluate VFA by CT, thus it is useful in the clinical setting.

4. In Fig 1, IMAC in severe group is higher in Obesity group 0. Is there difference in age between non-severe and severe groups in Obesity group 0?

5. In table 2, the authors show that IMAC is higher in severe group compared to non-severe group. However, in Fig 1, IMAC in group 1 and group 2-4 is not different between severe and non-severe groups. Could you elaborate the reason why there is no difference?

Reviewers' comments:

Reviewer's Responses to Questions

**Comments to the Author**

1. Is the manuscript technically sound, and do the data support the conclusions?

Reviewer #1: Partly

2. Has the statistical analysis been performed appropriately and rigorously? 

Reviewer #1: Yes

3. Have the authors made all data underlying the findings in their manuscript fully available?

Reviewer #1: No

4. Is the manuscript presented in an intelligible fashion and written in standard English?

Reviewer #1: No

5. Review Comments to the Author

Reviewer #1: I read the manuscript entitled “ Impact of body composition on patient prognosis after SARS-Cov-2 infection”. I think that this study is impressive, and important to understand the relationship between obesity/adiposity and the risk factors, prognosis and severity of COVID19 infection. I have some questions on the current manuscript.

I would like to refer to Methods and Study population. (Page11)

This study cohort is thought to be a retrospective observational study. I wonder if computed tomography (CT) scanning was performed on all the admitted patients with COVID19. Exclusion criteria from the study was equivocated or might not be written in the manuscript. The author should totally explain the study design, and illustrate a figure including the flowchart to sum up the method of recruiting and enrolling the patients and collecting their data.

I should mention about data collection in the study. (Page11-12)

The authors compared the differences of clinical parameters between the intubated patients and the non-intubated patients with COVID19. I wonder if there were patients with mild phenotype among the non-intubated patients, who did not need oxygen administration. I assume that most of the patients the author enrolled required oxygenation therapy: moderate type. I wonder if the data of patients without oxygenation (i.e., mild type) were involved. I thought that comparison between the patients with severe type and those with mild type is necessary.

The author should refer to the methods of measurement on the blood parameters, including equipment. The CT device they used should be precisely written.

The author should indicate the source of reference on the severity of COVID19 patients, the definition of obesity, and the classification of obesity. Did the author apply the guidelines for the Management of Obesity Disease 2022, which was published by Japanese Society of the Study of Obesity (JASSO) to the study design?

The data of IMAC are interesting. Figure 1(Page 28) indicates that ectopic adiposity in the muscle (i.e., myosteatosis) can affect the prognosis of COVID19 patients in non-obese population. I consider that myosteatosis can have a bigger influence on the severity of COVID19 infection compared with sarcopenia. What is its mechanism? The author may mention what they considered about that, even if it was speculation.

In the abstract (Page 7), the influence of SARS-CoV2 mutation for the severity of COVID19 does not matter in the context.

A native English speaker may correct grammatical problems in the current manuscript. And, the paper was organized again.

6. PLOS authors have the option to publish the peer review history of their article (what does this mean?). If published, this will include your full peer review and any attached files.

Reviewer #1: No

---

## [Author Response · Author response to Decision Letter 0]

6 Apr 2023

Professor Jun Mori

Academic Editor

PLOS ONE

PONE-D-23-01341

Impact of body composition on patient prognosis after SARS-Cov-2 infection

Dear Prof. Jun Mori

We appreciate your positive response and the opportunity to address the reviewer’s and editor’s comments regarding our submitted manuscript (PONE-D-23-01341). We would also like to take this opportunity to thank the reviewer for the time and insightful suggestions. We have modified our manuscript according to the reviewer’s and editor’s suggestions. As requested, a point by point response to each of the issues raised by the reviewer and editor now follows:

Reviewer #1

# I would like to refer to Methods and Study population. (Page11)

This study cohort is thought to be a retrospective observational study. I wonder if computed tomography (CT) scanning was performed on all the admitted patients with COVID19. Exclusion criteria from the study was equivocated or might not be written in the manuscript. The author should totally explain the study design, and illustrate a figure including the flowchart to sum up the method of recruiting and enrolling the patients and collecting their data.

We appreciate this comment. We basically performed CT scan for all the cases unless the patient had any problem for taking CT scan (for example, pregnancy), and patients who lacked CT images were excluded for further analysis. Totally, 93 patients were admitted in the study period. Of these, 17 patients were excluded from the study, for eight patients were pregnant and CT scans of abdomen were not performed, required area for anthropometric measurements were not captured in eight patients, and clear image was not obtained in one patient. This information was added in the Result section and we also added the flowchart of the patient enrollment as Figure 1. 

# I should mention about data collection in the study. (Page11-12)

The authors compared the differences of clinical parameters between the intubated patients and the non-intubated patients with COVID19. I wonder if there were patients with mild phenotype among the non-intubated patients, who did not need oxygen administration. I assume that most of the patients the author enrolled required oxygenation therapy: moderate type. I wonder if the data of patients without oxygenation (i.e., mild type) were involved. I thought that comparison between the patients with severe type and those with mild type is necessary.

 We agree this comment. However, all the patients admitted to our hospital had underlying disease and certain risks for disease severity of COVID-19, and hence all of the patients enrolled to this study needed oxygenation. Therefore, though we also think it is very important to compare the clinical and anthropometric parameters between severe and mild cases as the reviewer mentioned, we could not make this comparison for the reason described above.

# The author should refer to the methods of measurement on the blood parameters, including equipment. The CT device they used should be precisely written.

All the blood tests were performed at the Clinical Laboratory of Mie University Hospital using LABOSPECT 008 (Hitachi High-Tech Corporation, Tokyo, Japan) for blood chemistry, XN-3000 (Sysmex, Kobe, Japan) for complete blood count, and CS-5100 (Sysmex) for blood coagulation tests. CT scans were performed using Discovery CT750 HD (GE Health Care, Tokyo, Japan). We added this information to the Methods section of the text. 

# The author should indicate the source of reference on the severity of COVID19 patients, the definition of obesity, and the classification of obesity. Did the author apply the guidelines for the Management of Obesity Disease 2022, which was published by Japanese Society of the Study of Obesity (JASSO) to the study design?

　We apologize for the lack of the source of obesity classification. As the reviewer mentioned, we classified obese patients into four grades by BMI according to the Guidelines for the Management of Obesity Disease 2022 by Japanese Society for the Study of Obesity. We added this information to Methods section. In addition, we classified the patient who needed intubation and mechanical ventilation as severe cases. The indication for intubation and mechanical ventilation followed the “Clinical Management of Patients with COVID-19: A guide for front-line healthcare workers” by the Ministry of Health, Labour and Welfare, Japan. We also included this information in Method section.

# The data of IMAC are interesting. Figure 1(Page 28) indicates that ectopic adiposity in the muscle (i.e., myosteatosis) can affect the prognosis of COVID19 patients in non-obese population. I consider that myosteatosis can have a bigger influence on the severity of COVID19 infection compared with sarcopenia. What is its mechanism? The author may mention what they considered about that, even if it was speculation.

 We greatly appreciate this comment. The mechanism regarding the association between IMAC and disease severity of COVID-19 is still unclear. Furthermore, though it is possible that myosteatosis may have a bigger influence of the COVID-19 prognosis than sarcopenia as the reviewer suggested, we do not have enough evidences to fully explain this mechanism yet. However, we can speculate some explanations. Recent study by Yi et al. showed thoracic myosteatosis was related to progression of COVID-19, which is comparable with our results. They suggested that muscle adiposity causes systemic muscle weakness and diminished effort for coughing which is crucial for protection against respiratory infection, and leads to disease progression. We also agree their speculation. We can assume myosteatosis may have bigger effect on muscle weakness than sarcopenia. The other possible mechanism is the changes in myokines. It has been elucidated that skeletal muscle secrets several hundreds of myokines. Myokines have multiple effects on modulating immunological, metabolic and physiological activities of human body. We can speculate that muscle adiposity may change the myokine profiles and causes alterations in their metabolic or immunomodulating effects, which results in disease severity. However, these effects by myokines may become less remarkable as the patients get obese, because the effects of the larger amount of inflammatory adipokines secreted by visceral adipose tissue become more prominent. We added these descriptions to the Discussion section.

# In the abstract (Page 7), the influence of SARS-CoV2 mutation for the severity of COVID19 does not matter in the context.

 We agree this suggestion. Following this comment, we eliminated the descriptions regarding SARS-Cov-2 mutation from the text and rewrote conclusion in Abstract.

# A native English speaker may correct grammatical problems in the current manuscript. And, the paper was organized again.

 We have had our manuscript proofread by a professional English editing service to correct errors in the text. 

Additional Editor Comments:

The authors assess the body composition and the degree of sarcopenia between severe and non-severe patients with COVID-19. They concluded that Intra-muscular adipose tissue content (IMAC) could be a useful predictor of disease severity for patients with COVID-19. The data is somewhat interesting, but I have the following concerns.

1. Non-severe patients group includes young patient such as 20 years age, even though mean age was not different between severe and non-severe groups. In general, younger people tend to be less severe with COVID-19 infection compared to older people. I'm wondering why this young patient was admitted and CT scan was performed. You should show the patient's age for both groups in detail.

 In total, eight patients were in their 20s. Among them, 6 patients were pregnant and abdominal CT scans were not performed, therefore, they were excluded from the study. The remaining two patients were included to the study. One was 22 years old female who had obesity (BMI; 32.8) and T2DM, and the other was 26 years old male who had obesity (BMI; 37). Both of them became severe ill after admission and intubated and on mechanical ventilation. We have also reanalyzed the data and made a comparison of age between non-severe and severe group in each obesity grade. We will describe this result more in detail in the response to editor’s comment 4.

2. There are many risk factors of severity of COVID-19 other than obesity and T2DM, e.g. hypertension, hyperlipidemia, chronic lung disease, chronic kidney disease, tobacco habits. What are your thoughts on the involvement of these other risk factors?

 We think all those factors the editor mentioned are also very crucial for severity of COVID-19. However, most of the patients in this study were transferred from the other regional hospitals and many of them had not attended any medical institution regularly before COVID-19. Therefore, we regret to say we did not have enough patients’ information regarding those risk factors other than obesity and T2DM which could be assessed after admission to our hospital. 

3. Do you have the data of waist circumference, which is also an index of visceral fat mass? Measuring waist circumference is much easier to evaluate VFA by CT, thus it is useful in the clinical setting.

 We do not have the data of waist circumference. We agree measuring waist circumference is easier and less expensive way to assess visceral fat mass. However, it sometimes varies by the body posture or the measurers. Therefore, we used CT scan as a more precise and objective method to evaluate VFA in this study.

4. In Fig 1, IMAC in severe group is higher in Obesity group 0. Is there difference in age between non-severe and severe groups in Obesity group 0?

 We greatly appreciate this comment. To respond to this query, we made comparison of the age between non-severe and severe patients in each obesity group and found non-severe patients were younger than severe patients in obesity grade 0, while no difference of age was observed between non-severe and severe patients in obesity grade 1, 2-4 groups. As non-obese individuals are less likely to have other metabolic-associated risk factors, these results may suggest that high IMAC may be more crucial for COVID-19 severity in this group. In addition, we can also speculate non-obese younger patients had more favorable disease courses because they had less muscle adiposity compared to non-obese older patients. We added these results in Table 4 as a new Table for the revision version and our interpretation in the discussion. 

5. In table 2, the authors show that IMAC is higher in severe group compared to non-severe group. However, in Fig 1, IMAC in group 1 and group 2-4 is not different between severe and non-severe groups. Could you elaborate the reason why there is no difference?

 In the bottom of Table 2, it is shown that IMAC was not different between non-severe and severe groups. We think this result is consistent with Figure 1. 

Once again, we would like to thank you and the reviewer for your positive suggestions. We believe our manuscript has been much improved as a result of incorporation of these useful suggestions and hope you will now find it acceptable for publication in PLOS ONE.

Norihiko Yamamoto M.D., Ph.D.

Department of General Medicine

Mie University Hospital, 2-174 Edobashi, Tsu, Mie 514-8507, Japan

Tel: +81-59-231-5290

Fax: +81-59-232-5289

E-mail: kotetsu@med.mie-u.ac.jp

---

## [Decision Letter · Decision Letter 1]

2 May 2023

PONE-D-23-01341R1Impact of body composition on patient prognosis after SARS-Cov-2 infectionPLOS ONE

Dear Dr. Yamamoto,

Thank you for submitting your manuscript to PLOS ONE. After careful consideration, we feel that it has merit but does not fully meet PLOS ONE’s publication criteria as it currently stands. Therefore, we invite you to submit a revised version of the manuscript that addresses the points raised during the review process.

We look forward to receiving your revised manuscript.

Kind regards,

Jun Mori

Academic Editor

PLOS ONE

Journal Requirements:

Additional Editor Comments:

The authors responded to all my comments. I have no further concerns.

The authors should respond the additional reviewer's comments.

Reviewers' comments:

Reviewer's Responses to Questions

**Comments to the Author**

1. If the authors have adequately addressed your comments raised in a previous round of review and you feel that this manuscript is now acceptable for publication, you may indicate that here to bypass the “Comments to the Author” section, enter your conflict of interest statement in the “Confidential to Editor” section, and submit your "Accept" recommendation.

Reviewer #1: All comments have been addressed

2. Is the manuscript technically sound, and do the data support the conclusions?

Reviewer #1: Yes

3. Has the statistical analysis been performed appropriately and rigorously? 

Reviewer #1: Yes

4. Have the authors made all data underlying the findings in their manuscript fully available?

Reviewer #1: Yes

5. Is the manuscript presented in an intelligible fashion and written in standard English?

Reviewer #1: Yes

6. Review Comments to the Author

Reviewer #1: I must refer to the discussion.

The author must precisely read the references you used, especially Ref.31 written by Bay ML. He precisely described that anti-inflammatory effects are induced by the muscle through physical exercise.

The author inserted new discussion, using the review article ( Ref.31). That article shows many kinds of myokine induced by physical exercise. Unfortunately, author’s description in the discussion was equivocal.

I understand that the muscle-derived (md) IL-6 increases by exercise, and mdIL6 suppresses TNF-alfa production, through that review article. Certainly, mdIL6 is anti-inflammatory cytokine. Exercise may be protective to inflammation, or COVID19 pneumonia. This context that the author wrote would be correct.

However, I wonder if intramuscular hyperadiposity, which means high IMAC in the muscle, could lead to inflammatory excitement in the lung or immune system in patients with COVID19 pneumonia. This discussion matters.

Moreover, I would like to know whether intramuscular adipocyte specifically secrets inflammatory cytokines and adipokines. Or I would like to know whether there could be crosstalk between the muscle and the adipocytes in muscle in immune response. Such discussion is of importance, and should be added, and be written concretely. Discussion should be more profound and meaningful.

7. PLOS authors have the option to publish the peer review history of their article (what does this mean?). If published, this will include your full peer review and any attached files.

Reviewer #1: No

---

## [Author Response · Author response to Decision Letter 1]

10 Jun 2023

Professor Jun Mori

Academic Editor

PLOS ONE

PONE-D-23-01341R1

Impact of body composition on patient prognosis after SARS-Cov-2 infection

Dear Prof. Jun Mori

We appreciate your positive response and the opportunity to address the reviewer’s and editor’s comments regarding our submitted manuscript (PONE-D-23-01341R1). We would also like to take this opportunity to thank the reviewer for the time and insightful suggestions. We have modified our manuscript according to the reviewer’s suggestions. As requested, a point by point response to the issue raised by the reviewer now follows:

Reviewer #1: I must refer to the discussion.

The author must precisely read the references you used, especially Ref.31 written by Bay ML. He precisely described that anti-inflammatory effects are induced by the muscle through physical exercise.

The author inserted new discussion, using the review article (Ref.31). That article shows many kinds of myokine induced by physical exercise. Unfortunately, author’s description in the discussion was equivocal.

I understand that the muscle-derived (md) IL-6 increases by exercise, and mdIL6 suppresses TNF-alfa production, through that review article. Certainly, mdIL6 is anti-inflammatory cytokine. Exercise may be protective to inflammation, or COVID19 pneumonia. This context that the author wrote would be correct.

However, I wonder if intramuscular hyperadiposity, which means high IMAC in the muscle, could lead to inflammatory excitement in the lung or immune system in patients with COVID19 pneumonia. This discussion matters.

Moreover, I would like to know whether intramuscular adipocyte specifically secrets inflammatory cytokines and adipokines. Or I would like to know whether there could be crosstalk between the muscle and the adipocytes in muscle in immune response. Such discussion is of importance, and should be added, and be written concretely. Discussion should be more profound and meaningful.

We greatly appreciate this comment. We agree our discussion about myokines is equivocal, therefore, we focused on IL-6 in the revised version, for it is one of the most important and well-studied inflammatory myokine. It is reported that increased level of IL-6 is produced by the muscle tissues with fatty regeneration (Addison O, et al.). As the reviewer pointed out, muscle-derived IL-6 has some anti-inflammatory effect. However, IL-6 has two-way signal transductions by the differed receptors. One signaling (classic signaling) induces generative and protective responses to small subset of cells (hepatocytes and leucocytes), the other signaling (trans-signaling) induces proinflammatory responses to all cells in the body. Moreover, it is reported that IL-6 plays a crucial role for the pathogenesis of cytokine release syndrome which leads disease severity of COVID-19. In muscle cells, though IL-6 has some beneficial effects for muscle growth, myogenesis or energy metabolism, it also has negative effects such as inducing muscle atrophy or wasting that could affect COVID-19 prognosis.

We added these things concisely and modified discussion section to make it less equivocal and more profound. We also replaced Ref 31 to another references (Ref. 31, 32, 33) which more adequately support our description. 

Once again, we would like to thank you and the reviewer for the positive suggestion. We believe our manuscript has been further improved as a result of incorporation of the reviewer’s suggestion and hope you will now find it suitable for publication in PLOS ONE.

---

## [Decision Letter · Decision Letter 2]

25 Jun 2023

PONE-D-23-01341R2Impact of body composition on patient prognosis after SARS-Cov-2 infectionPLOS ONE

Dear Dr. Yamamoto,

Thank you for submitting your manuscript to PLOS ONE. After careful consideration, we feel that it has merit but does not fully meet PLOS ONE’s publication criteria as it currently stands. Therefore, we invite you to submit a revised version of the manuscript that addresses the points raised during the review process.

We look forward to receiving your revised manuscript.

Kind regards,

Jun Mori

Academic Editor

PLOS ONE

Journal Requirements:

Additional Editor Comments:

All comments have been addressed.

Reviewers' comments:

Reviewer's Responses to Questions

**Comments to the Author**

1. If the authors have adequately addressed your comments raised in a previous round of review and you feel that this manuscript is now acceptable for publication, you may indicate that here to bypass the “Comments to the Author” section, enter your conflict of interest statement in the “Confidential to Editor” section, and submit your "Accept" recommendation.

Reviewer #1: All comments have been addressed

2. Is the manuscript technically sound, and do the data support the conclusions?

Reviewer #1: Partly

3. Has the statistical analysis been performed appropriately and rigorously? 

Reviewer #1: Yes

4. Have the authors made all data underlying the findings in their manuscript fully available?

Reviewer #1: Yes

5. Is the manuscript presented in an intelligible fashion and written in standard English?

Reviewer #1: Yes

6. Review Comments to the Author

Reviewer #1: Obesity can also predict the severity of COVID19 pneumonia in Japan as well as in the United States and European countries. In Japan, the population of severe obesity is fewer than in the United States and European countries. The current study suggests that IMAC might be predictive for the severity of COVID19 pneumonia in non-obese Japanese patients. In that context, to discover a predictor on the severity of COVID19 pneumonia is greatly meaningful in Japan. The finding that the author discovered on IMAC in this study is interesting to me.

However, IMAC does not look different in the Japanese obese patients with COVID19 pneumonia. So, IMAC may not be useful to predict the severity of COVID19 pneumonia as well as mortality in Japan.

The author should modify the conclusion in the abstract in order to avoid misunderstanding.

7. PLOS authors have the option to publish the peer review history of their article (what does this mean?). If published, this will include your full peer review and any attached files.

Reviewer #1: No

---

## [Author Response · Author response to Decision Letter 2]

29 Jun 2023

Professor Jun Mori

Academic Editor

PLOS ONE

PONE-D-23-01341R2

Impact of body composition on patient prognosis after SARS-Cov-2 infection

Dear Prof. Jun Mori

We appreciate your positive response and the opportunity to address the reviewer’s and editor’s comments regarding our submitted manuscript (PONE-D-23-01341R2). We would also like to take this opportunity to thank the reviewer for the time and insightful suggestions. We have modified our manuscript according to the reviewer’s suggestions. As requested, a point by point response to the issue raised by the reviewer now follows:

Reviewer #1:

 Obesity can also predict the severity of COVID19 pneumonia in Japan as well as in the United States and European countries. In Japan, the population of severe obesity is fewer than in the United States and European countries. The current study suggests that IMAC might be predictive for the severity of COVID19 pneumonia in non-obese Japanese patients. In that context, to discover a predictor on the severity of COVID19 pneumonia is greatly meaningful in Japan. The finding that the author discovered on IMAC in this study is interesting to me.

However, IMAC does not look different in the Japanese obese patients with COVID19 pneumonia. So, IMAC may not be useful to predict the severity of COVID19 pneumonia as well as mortality in Japan.

The author should modify the conclusion in the abstract in order to avoid misunderstanding.

We agree this comment. Our description in the Abstract may cause misunderstanding. As the reviewer stated, our data showed IMAC was different between severe and non-severe COVID-19 cases in non-obese patients. And it does not predict disease course in obese patients or disease mortality in any obesity grade. We modified the conclusion in the Abstract to make these points clearer according to the reviewer’s suggestion.

Once again, we would like to thank you and the reviewer for the positive suggestion. We believe our manuscript has been further improved as a result of incorporation of the reviewer’s suggestion and hope you find it meet PLOS ONE’s publication criteria now.

Norihiko Yamamoto M.D., Ph.D.

Department of General Medicine

Mie University Hospital, 2-174 Edobashi, Tsu, Mie 514-8507, Japan

Tel: +81-59-231-5290

Fax: +81-59-232-5289

---

## [Decision Letter · Decision Letter 3]

14 Jul 2023

Impact of body composition on patient prognosis after SARS-Cov-2 infection

PONE-D-23-01341R3

Dear Dr. Yamamoto,

We’re pleased to inform you that your manuscript has been judged scientifically suitable for publication and will be formally accepted for publication once it meets all outstanding technical requirements.

Kind regards,

Jun Mori

Academic Editor

PLOS ONE

Additional Editor Comments (optional):

The authors responded to the reviewer's comments correctly.

Reviewers' comments:

Reviewer's Responses to Questions

**Comments to the Author**

1. If the authors have adequately addressed your comments raised in a previous round of review and you feel that this manuscript is now acceptable for publication, you may indicate that here to bypass the “Comments to the Author” section, enter your conflict of interest statement in the “Confidential to Editor” section, and submit your "Accept" recommendation.

Reviewer #1: All comments have been addressed

2. Is the manuscript technically sound, and do the data support the conclusions?

Reviewer #1: Yes

3. Has the statistical analysis been performed appropriately and rigorously? 

Reviewer #1: Yes

4. Have the authors made all data underlying the findings in their manuscript fully available?

Reviewer #1: Yes

5. Is the manuscript presented in an intelligible fashion and written in standard English?

Reviewer #1: Yes

6. Review Comments to the Author

Reviewer #1: The modificaton the author has done is acceptable to me.

It is worth publishing the current manuscript.

7. PLOS authors have the option to publish the peer review history of their article (what does this mean?). If published, this will include your full peer review and any attached files.

Reviewer #1: No

---

## [Editor Report · Acceptance letter]

20 Jul 2023

PONE-D-23-01341R3 

Impact of body composition on patient prognosis after SARS-Cov-2 infection 

Dear Dr. Yamamoto:

I'm pleased to inform you that your manuscript has been deemed suitable for publication in PLOS ONE. Congratulations! Your manuscript is now with our production department. 

Kind regards, 

on behalf of

Dr. Jun Mori 

Academic Editor

PLOS ONE